# Model Informed Precision Dosing Tool Forecasts Trough Infliximab and Associates with Disease Status and Tumor Necrosis Factor-Alpha Levels of Inflammatory Bowel Diseases

**DOI:** 10.3390/jcm11123316

**Published:** 2022-06-09

**Authors:** Christian Primas, Walter Reinisch, John C. Panetta, Alexander Eser, Diane R. Mould, Thierry Dervieux

**Affiliations:** 1Division of Gastroenterology & Hepatology, Medical University of Vienna, A-1090 Vienna, Austria; christian.primas@meduniwien.ac.at (C.P.); dr.eser@gmx.at (A.E.); 2St. Jude Children’s Research Hospital, Memphis, TN 38105, USA; carl.panetta@stjude.org; 3Projection Research LLC, Fort Myers, FL 33901, USA; drmould@pri-home.net; 4Prometheus Laboratories, San Diego, CA 92121, USA

**Keywords:** Infliximab, therapeutic drug monitoring, model informed precision dosing, inflammatory bowel disease

## Abstract

Background: Substantial inter-and intra-individual variability of Infliximab (IFX) pharmacokinetics necessitates tailored dosing approaches. Here, we evaluated the performances of a Model Informed Precision Dosing (MIPD) Tool in forecasting trough Infliximab (IFX) levels in association with disease status and circulating TNF-α in patients with Inflammatory Bowel Diseases (IBD). Methods: Consented patients undergoing every 8-week maintenance therapy with IFX were enrolled. Midcycle specimens were collected, IFX, antibodies to IFX, albumin were determined and analyzed with weight using nonlinear mixed effect models coupled with Bayesian data assimilation to forecast trough levels. Accuracy of forecasted as compared to observed trough IFX levels were evaluated using Demings’s regression. Association between IFX levels, CRP-based clinical remission and TNF-α levels were analyzed using logistic regression and linear mixed effect models. Results: In 41 patients receiving IFX (median dose = 5.3 mg/Kg), median IFX levels decreased from 13.0 to 3.9 µg/mL from mid to end of cycle time points, respectively. Midcycle IFX levels forecasted trough with Deming’s slope = 0.90 and R2 = 0.87. Observed end cycle and forecasted trough levels above 5 µg/mL associated with CRP-based clinical remission (OR = 7.2 CI95%: 1.7–30.2; OR = 21.0 CI95%: 3.4–127.9, respectively) (*p* < 0.01). Median TNF-α levels increased from 4.6 to 8.0 pg/mL from mid to end of cycle time points, respectively (*p* < 0.01). CRP and TNF-α levels associated independently and additively to decreased IFX levels (*p* < 0.01). Conclusions: These data establish the value of our MIPD tool in forecasting trough IFX levels in patients with IBD. Serum TNF-α and CRP are reflective of inflammatory burden which impacts exposure.

## 1. Introduction

Therapeutic drug monitoring (TDM) assists gastroenterologists with the management of Inflammatory Bowel Diseases (IBD; Crohn’s Disease [CD] and Ulcerative Colitis [UC]) by clinical pharmacokinetic (PK) assessment of Infliximab (IFX) and antibodies to Infliximab (ATI) levels to detect underexposure and to prevent negative outcomes. The American Gastroenterological association has endorsed the TDM of IFX, and maintenance IFX trough threshold of 5 µg/mL was proposed as an effective minimum target level that can maximize TNF-α neutralization capabilities of the drug to promote inflammatory control of the underlying disease [1,2]. Reactively, in the face of uncontrolled disease, TDM can be implemented to inform on the value of dose intensification in the presence of inadequate exposure and accelerated clearance of the drug, while proactively, sustained maintenance of IFX levels to promote drug tolerance can help avoid the re-emergence of inflammation and flare post induction [3,4].

In recent years, modern and sophisticated Model Informed Precision Dosing (MIPD) tools that employ clinical PK have emerged to guide IFX dosing [5], and both retrospective and prospective clinical utility studies support the value of such dosing optimization to improve outcome [6,7]. Ideally, the optimization of IFX dose during maintenance should be proactively based on the identification of patients likely to present with underexposure at the end of infusion cycle, and as such, the collection of specimens during the elimination phase of IFX to forecast trough levels may inform clinicians of impending risk for underexposure that could be addressed by dose intensification.

Inflammatory burden remains a key determinant of IFX exposure; circulating C-reactive Protein (CRP) and fecal calprotectin are commonly measured to assist clinicians to assess underlying inflammation. However, despite the fact that IFX and anti-TNF blockers have been used in IBD and other immune mediated diseases for decades, the association between antigenic TNF-α and IFX PK is very limited, although high levels of TNF-α at baseline or during treatment serves as a sink for IFX thereby promoting underexposure, inadequate disease control, as reported previously in IBD [8] and rheumatoid arthritis [9].

The study establishing the validity of MIPD in forecasting trough concentration and associating with disease control is needed before implementation in clinical practice and the clinical laboratory setting. In this report, we establish the performance characteristics of MIDP in forecasting trough IFX levels and in associating these levels with disease control. We also evaluated the changes in antigenic TNF-α levels during infusion cycle and the impact on IFX exposure.

## 2. Materials and Methods

### 2.1. Patients and Laboratory Measurements

Consented patients with IBD (CD and UC) undergoing every 8-week maintenance therapy with IFX were enrolled at a single site. Two specimens were collected within one maintenance cycle, a first specimen collected mid-cycle at least 20 days after infusion, and a second specimen collected towards the end of cycle. Serum was isolated from the clot immediately after specimen collection and stored at −20 °C until analysis. IFX levels (assay range 0.8–34 µg/mL) and Antibodies to Infliximab (ATI) (cutoff > 3.1 U/mL for positive status) were determined from serum using drug tolerant homogenous mobility shift assay as described previously [10] (Prometheus Laboratories, San Diego, CA, USA). Serum albumin (g/L) and CRP (mg/L) were measured using standard immunoassay techniques (IMMAGE® 800 Protein Chemistry Analyzer, Beckman Coulter, Brea, CA, USA) while antigenic TNF-α levels (following one freeze thaw cycle) were measured using high sensitivity immunoassays (Singulex Erenna Assay, MilliporeSigma, Burlington, MA, USA) and expressed as pg/mL [11]. Disease activity was assessed using Harvey-Bradshaw Index and partial Mayo score for CD and UC patients, respectively. Outcome variable consisted of clinical and biochemical remission (HBI and Partial Mayo below 5 and 2 points, respectively, with CRP level below 3 mg/L).

### 2.2. Model Informed Precision Dosing Tool

Individual PK parameters were estimated using a combinations of nonlinear mixed effect models (Monolix 2020R1, Lixoft, Paris, France) coupled with R functions (version 4.0) translated from MatLab (R2021a) and prior information from previously reported population pharmacokinetics model [7] independently of the patients enrolled in the study. The model employed two compartment pharmacokinetics with random effects on clearance (Cl), volume of distribution (central, [V1] and peripheral [V2]) and intercompartment clearance (Q). Covariates consisted of weight (on Cl, V1, Q and V2), Albumin (on Cl) and positive ATI status (on Cl). All parameters were fixed as described [7], with the exception that the proportional residual error model was set at 0.10. For each subject, mid-cycle IFX levels, ATI status, albumin and weight were used to estimate the conditional distribution of the individual parameters, which represents the uncertainty of the individual parameters given the observations collected above, and the prior information [7]. Conditional distributions of the model parameters (clearance (Cl), central volume of distribution (V1), intercompartmental clearance (Q), and peripheral volume of distribution (V2)) were generated for each patient using Markov Chain Monte Carlo simulations (Metropolis Hastings algorithm), and sampling (n = 100) from those distributions were used to estimate the median forecasted end of cycle, median trough levels (immediately before infusion, 56 days post infusion) and median probability to achieve trough levels above 5 and 10 µg/mL. Prediction intervals (80% corresponding to the 10th and 90th percentile of the estimates levels) were also calculated. All observations below the limit of quantitation of the IFX assay (0.8 µg/mL) were censored.

### 2.3. Statistical Analysis

Performances of the MIPD in forecasting end of cycle IFX levels using mid-cycle determination was assessed using Deming regression, regression coefficient and Kappa statistics (at 5 µg/mL cutoff). Area under the Receiver Operating Characteristic curve (AUC ROC) and logistic regression were used to evaluate the association of individual parameters with outcome measure. Group comparisons were tested using Kruskal Wallis ANOVA while longitudinal changes in CRP and TNF-α levels in relation to IFX exposure were evaluated using linear mixed effect models.

## 3. Results

A total of 41 patients with IBD (31 with CD and 10 with UC) undergoing every 8-week maintenance therapy with IFX (median dose = 5.9 mg/Kg [interquartile range, IQR:5.3–6.2]) were enrolled. Mid-cycle specimens were collected 28-day (median) post infusion (interquartile range [IQR]: 26–30-days), while end of cycle were collected 52-days (median) post infusion (IQR: 44–56-days). Patient characteristics and laboratory measures are presented in Table 1, 44% (18/41) patients presented with inactive disease (biochemical and clinical remission). Montreal classification criteria is provided in Table 2. Median individual PK parameter estimated using mid-cycle determination yielded 0.300 L/day CL (IQR 0.240–0.424), with 3.36 (IQR: 3.03–3.70) and 1.56 L (IQR: 1.37–1.77) for V1 and V2, respectively. Median Q was 0.134 L/day (IQR: 0.108–0.153). There was no significant difference in IFX Clearance between the group of patients who received concomitant immunomodulators (median 0.323 L/day, IQR: 0.295–0.385, n = 8) as compared to those who did not (median 0.301 L/day, IQR: 0.234–0.435, n = 33) (*p* = 0.68).

### 3.1. End of Cycle IFX Levels Can Be Forecasted Using Mid Cycle Determinations

A typical pK profile of patient under IFX is presented in Figure 1.

The performance characteristics of the Bayesian method in forecasting end of cycle levels using mid-cycle determinations is presented in Figure 2. Median forecasted end of cycle IFX levels (at the time of specimen collection) were 4.2 µg/mL (IQR: 2.2–8.3 µg/mL) and yielded Demings slope of 0.90 CI95%: 0.78 to 1.02) with 0.87 regression coefficient (R^2^). We also tested the performances of the MIPD in the group of patients with (R^2^ = 0.89; slope = 0.91; n = 18) or without clinical and biochemical remission status achieved (R^2^ = 0.75; slope = 0.92; n = 23). A total of 17/41 (41%) patient specimens presented with IFX levels greater than 5 µg/mL at end of cycle, of which 14 specimens were predicted by the forecasting method as being greater than 5 µg/mL (82%). Alternatively, 24/41 (59%) patients specimens presented with end of cycle levels below 5 µg/mL, of which 21 specimens were predicted by the forecasting method as being below 5 µg/mL (87.5%). Kappa statistics at cutoff of 5 µg/mL was 0.70 ± 0.11. Forecasted median trough IFX levels (at 56 days) were 3.0 µg/mL (IQR: 1.6–7.6); individualized probability to achieve Trough IFX levels above 5 µg/mL was 0.21 (median, IQR: 0.07–0.78). The forecast time to reach threshold within maintenance cycle was 48 days (median, IQR: 37–68 days). The probability to achieve trough levels above 10 µg/mL was very low in this cohort (median 0.05 IQR: 0.02–0.21).

### 3.2. Forecasted IFX Levels Associate with Clinical and Biochemical Remission

The association between observed end of cycle and forecasted trough IFX levels with clinical and biochemical validation was tested in the 41 patients. Patients presenting with observed end of cycle IFX levels above 5 µg/mL were 7.2-fold (CI95%: 1.7–30.2) more likely to present with clinical and biochemical remission (*p* < 0.001) as compared to patients presenting with suboptimal exposure (below 5 µg/mL) (*p* < 0.01). Similarly, forecasted trough IFX levels above 5 µg/mL associated with clinical and biochemical remission (OR = 21.0 CI95%: 3.4–127.9) (*p* < 0.001). Results are summarized in Table 3.

As presented in Figure 3, the AUC under the ROC with clinical and biochemical remission was comparable between observed end of cycle IFX (AUC = 0.778; CI95%: 0.626–0.929) and forecasted trough (AUC= 0.766; CI95%: 0.599–0.933). The probability to achieve trough levels above target threshold of 5 µg/mL yielded an AUC of 0.761 (CI95%: 0.590–0.931) (OR range = 37.4; CI95%: 3.6–385.5) (*p* < 0.001). Median time to reach threshold exposure was 40.5 days (IQR: 28–50.5 days) among patients with active disease as compared to 65.5 days (IQR: 50–77) among patients with biochemical and clinical remission (*p* < 0.01).

### 3.3. Circulating TNF-α and CRP Levels Independently and Additively Impact IFX Clearance in IBD

There was significant increase in TNF-α levels from midcycle to end cycle (*p* < 0.01) (Table 1). In contrast the change in CRP level was not significant (*p* = 0.44). There was a directional trend between higher TNF-α levels and active disease that did not reach significance (*p* > 0.10; data not shown). Linear mixed effect models revealed that the change in TNF-α were independent of those from CRP (*p* = 0.67). In contrast the change in IFX levels from midcycle to end of cycle associated independently with CRP (marginal R^2^ = 0.06; *p* = 0.047), TNF-α (marginal R2 = 0.21; *p* < 0.01). Multivariate analysis indicated that the impact of those inflammatory markers on IFX were additive (marginal R^2^ = 0.27). Results are summarized in Table 4.

Finally, multivariate logistic regression analysis revealed that patients presenting with forecasted trough IFX levels above 5 µg/mL were 13.9-fold (adjusted OR, CI95%: 2.0–94.9; *p* < 0.01) and 7.3-fold (adjusted OR, CI95%: 1.3–40.4; *p* < 0.01) less likely to have end cycle CRP levels above 3 mg/L, and TNF-α above 8 pg/mL (median), respectively. The cumulative impact of the inflammatory burden on IFX exposure is presented in Table 5.

## 4. Discussion

There is considerable interpatient variability in drug exposure following standard dosing of monoclonal antibodies such as IFX, and MIDP tools are poised to improved patient outcome by guiding dose and improving disease control [7,12]. These MIDP tools typically employ Bayesian forecasting methods for dose individualization and have been implemented for several therapies such as busulfan [13], vancomycin [14] and may be also helpful for controlling side effects in oncology [15]. In this report we have established the performance characteristics of MIPD in forecasting trough IFX concentrations by using prior information from a model that uses two compartment pharmacokinetics with weight, ATI and albumin as covariates [7]. In this study, the parameter estimates from the prior published models were tested in this validation cohort with specimens collected during the linear phase of elimination of IFX, and trough levels were forecasted by calculating the conditional distribution of individual parameters. Traditionally MIPD rely on the estimation of maximum a posteriori but as reported previously these methods provide limited information and no do not reflect the uncertainty of the measurement for decision making [15,16]. In this report, Bayesian assimilation techniques were employed by calculating the probability to achieve a certain pre-specified trough.

The forecasting method revealed generally good agreement between forecasted and observed end of cycle levels and thus suggests that collection of one specimen during mid cycle may be sufficient to ascertain and forecast of subsequent trough levels. While this validation cohort establishes the performances of the MIPD tool in fore-casting trough the number of patients enrolled was limited and additional data are currently collected to confirm these findings.

There are several clinical applications with the MIPD tool. First, the collection of patient specimen during the linear terminal phase of elimination of IFX may help identify early those patients who are likely to present with suboptimal exposure at trough and thus facilitate the implementation of countermeasures during this window of opportunity (i.e., dose intensification at the next scheduled dose). Second the MIPD tool can allow the most appropriate time for dosing where IFX levels reach suboptimal threshold and can inform on dose and dose interval combinations that can produce desired trough levels commensurate with target troughs (i.e., >5 or 10 µg/mL). While endoscopic assessments were not available in this study, we established a significant association between forecasted trough IFX levels and clinical and biochemical remission, and these data add the already existing large body of evidence supporting the value of PK measurements in association with disease control [1,4]. We also reported a significant association between the disease outcome variable with the probability of achieving trough concentration above 5 µg/mL. However, a very low number of patients achieved forecasted trough concentration above 10 µg/mL and thus suggest the opportunity to improve outcome by dosing intensification in a significant proportion of patients. These data illustrate the potential of MIPD in optimizing Infliximab therapy, and this premise could also be applied to other monoclonal antibodies such as adalimumab that is also prone to underexposure leading to poor disease control.

While anti-TNF therapy have been available for over two decades, there is paucity of data reporting the impact of antigenic inflammatory TNF-α levels on IFX exposure in IBD, and clinicians have traditionally relied on serum CRP as indicator of inflammation. This is primarily due to past constraints in that there were several limitations with the determination of TNF-α levels in the clinical practice setting, owing to significant pre-analytical variations associated with the stability of TNF-α during specimen transportation and processing and the challenge in determining pg quantities of the cytokine. In this report we used highly sensitive immunoassay to quantify TNF-α levels and our data suggest a strong association between IFX and TNF-α within maintenance cycle, whereby the decrease in IFX levels during the elimination phase parallels a rise in circulating TNF-α. Recent studies using drug tolerant assays have established that treatment with adalimumab associates with an increase TNF-α, most likely reflecting the formation of adalimumab and TNF-α complexes18. In our study, only free TNF-α levels were determined and thus it is not surprising that the decrease IFX levels associated with lower TNF-α neutralization and thus increase in free circulating levels of the cytokine. Whether these changes in the circulation also associate with tissue dynamics in inflamed gut is not known, but likely, and we speculate that inadequate exposure (as seen in most patients presenting with suboptimal IFX trough levels) may insufficiently provide TNF-α neutralizing capabilities as seen in patients with rheumatoid arthritis [9]. However, we acknowledge that perhaps due to the small size we were not able to detect a significant association between TNF-α levels and disease activity status in contrast to other studies and the levels detected in our assay may not be biologically active TNF-α [9,17,18].

Interestingly there was also no significant correlation between CRP and TNF-α and our analysis revealed that both CRP and TNF-α contributed independently and additively to IFX exposure. CRP is an acute phase reactant protein produced by the liver and most likely reflected the dynamics of inflammatory cytokine other than TNF-α such as Il-6 and IL-1 [19]. It follows that those multiple pathways can impact IFX exposure, and as already proposed, this inflammatory burden may sink IFX levels below threshold levels commensurate with disease control [8].

In conclusion, our data help support the implementation of MIPD to forecast IFX exposure in IBD and provide insights into the complex dynamics between IFX and antigenic TNF-α in IBD.

## Figures and Tables

**Figure 1 jcm-11-03316-f001:**
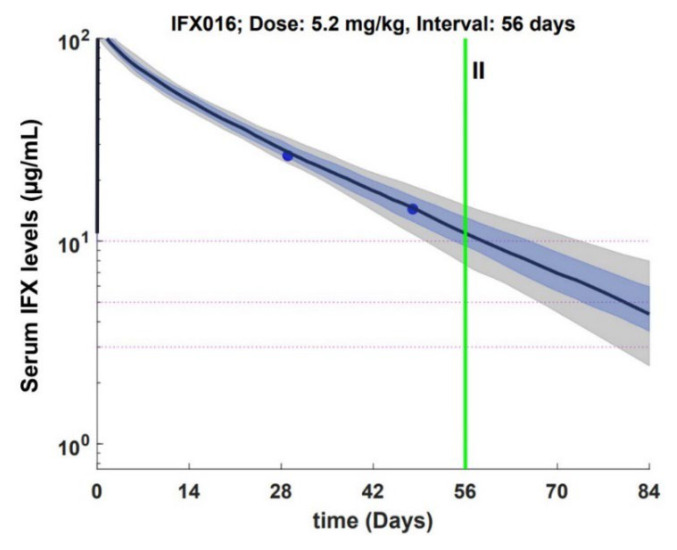
Performance of MIPD in forecasting Trough levels, PK profile. Solid line corresponds to the median IFX levels calculated from 100 random samples from the condition distribution of individual Pk parameters. Blue zone corresponds to interquartile range, grey zone corresponds to the 10th and 90th percentile.

**Figure 2 jcm-11-03316-f002:**
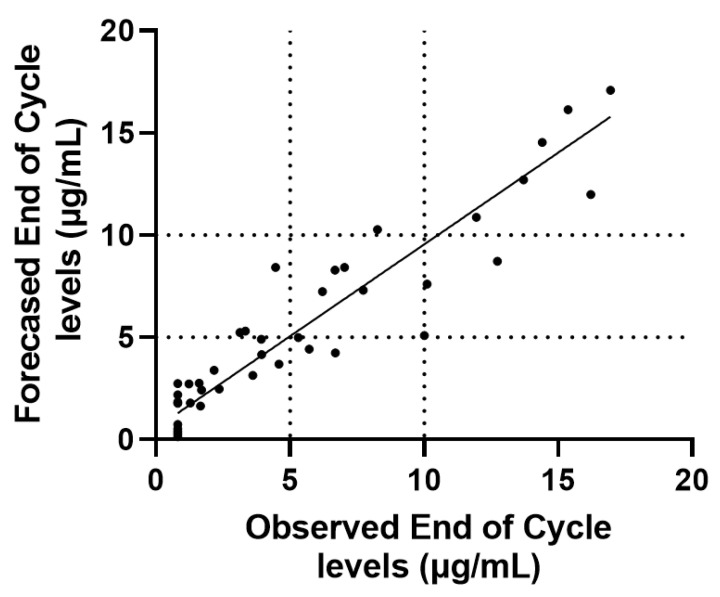
Comparison between observed and forecasted end of cycle IFX levels (Deming’s slope = 0.90; slope = 0.87).

**Figure 3 jcm-11-03316-f003:**
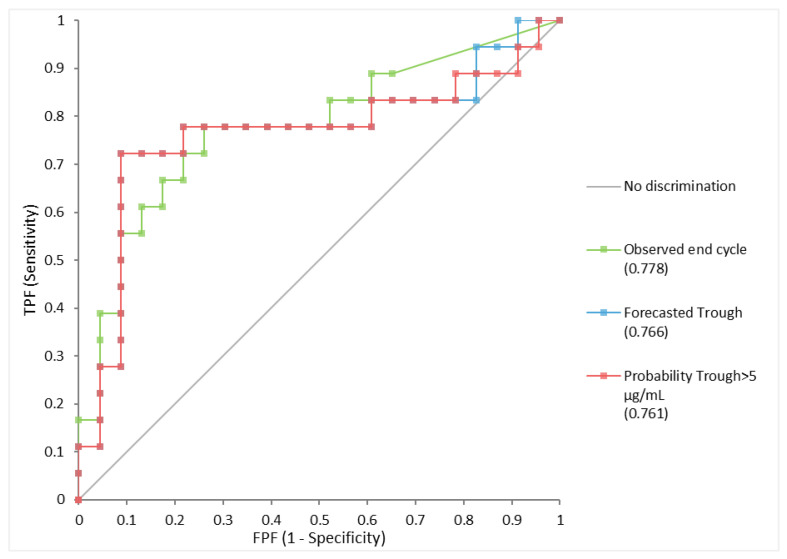
**ROC of Observed and Forecasted IFX levels with clinical and biochemical remission.** Measured endcycle IFX levels: AUC = 0.778 (CI95%: 0.626–0.929); Forecasted trough levels: AUC= 0.766 (CI95%: 0.599–0.936) (*p* = 0.89); Forecasted probability of trough level > 5 µg/mL: AUC= 0.756 (CI95%: 0.586–0.926).

**Table 1 jcm-11-03316-t001:** Patient characteristics. Results are expressed as median (IQR), as appropriate.

	Estimate
Gender (female)	46.3% (19/41)
Age (years)	43 (34–49)
CD Diagnosis (vs UC)	76% (31/41)
Concomitant Immunomodulators	20% (8/41)
Albumin (g/L)	42.6 (40.4–44.3)
IFX (µg/mL)	
Mid cycle	13.2 (6.2–20.5)
End cycle	3.9 (1.2–7.7)
ATI status	
Mid cycle	15% (6/41)
End cycle	19% (8/41)
CRP (mg/L)	
Mid cycle	2.4 (0.6–5.4)
End cycle	2.7 (0.1–8.8)
TNF-α (pg/mL)	
Mid cycle	4.6 (3.1–8.8)
End cycle	8.0 (4.8–12.0)
Disease activity	
HBI	2 (1–5)
Partial Mayo	0 (0–3)
Clinical & Biochemical Remission	44% (18/41)

**Table 2 jcm-11-03316-t002:** Montreal Classification criteria.

	**n/N**
**Crohn’s Disease**	
**Age**:	
≤16 (A1)	2/31
17–40 (A2)	25/31
>40 (A3)	4/31
**Location**	
Terminal Ileum (L1)	2/31
Colon (L2)	6/31
Ileocolon (L3)	23/31
**Behavior**	
Non structuring, non-penetrating (B1)	15/31
Stricturing (B2)	12/31
Penetrating (B3)	3/31
Perianal	1/31
**Ulcerative Colitis**	
Proctitis (E1)	0/10
Left sided (E2)	3/10
Pancolitis (E3)	7/10

**Table 3 jcm-11-03316-t003:** Performance of Observed and Forecasted IFX levels in associating with clinical and biochemical remission.

	Observed End of Cycle > 5 µg/mL			Forecasted Trough > 5 µg/mL		
	Estimate	−95CI	+95CI	Estimate	−95CI	+95CI
Sensitivity	0.67	0.44	0.84	0.67	0.44	0.84
Specificity	0.78	0.58	0.90	0.91	0.73	0.98
False positive	0.22	0.10	0.42	0.09	0.02	0.27
False Negative	0.33	0.16	0.56	0.33	0.16	0.56
Positive LR	3.1	1.4	7.2	7.7	2.3	28.4
Negative LR	0.4	0.2	0.8	0.36	0.2	0.6
Odds Ratio	7.2	1.8	30.2	21.0	3.9	109.0

LR: likelihood ratio.

**Table 4 jcm-11-03316-t004:** Linear Mixed effect Model of IFX levels in relation to inflammatory markers. Each unit increase in TNF-α associated with 0.33 ± 0.08 µg/mL decrease in IFX levels.

	Intercept	Slope ± SEM	*p* Value	Marginal R^2^
**CRP**	10.0 ±1.2	−0.13 ± 0.07	0.047	0.058
**TNF-α**	12.4 ± 1.2	−0.33 ± 0.08	<0.001	0.213
**CRP + TNF-α**	13.4 ± 1.3	−0.14 ± 0.06 −0.33 ± 0.07	0.022 <0.001	0.271

**Table 5 jcm-11-03316-t005:** IFX exposure in relation to Inflammatory burden.

End of Cycle	CRP ≤ 3 mg/L and TNF-α ≤ 8 pg/mL	CRP > 3 mg/L or TNF-α > 8 pg/mL	CRP > 3 mg/L and TNF-α > 8 pg/mL	*p* Value
Observed end cycle µg/mL	7.0 (4.6–14.4)	3.3 (0.8–7.7)	1.7 (0.8–3.9)	<0.01
Forecasted Trough µg/mL	6.7 (5.3–9.0)	3.7 (0.7–7.7)	2.0 (1.1–3.2)	<0.01
Probability Trough > 5 µg/mL	0.68 (0.47–0.90)	0.25 (0.05–0.70)	0.10 (0.04–0.19)	<0.01

## Data Availability

No data sharing.

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
