# Peer review of "Model Informed Precision Dosing Tool Forecasts Trough Infliximab and Associates with Disease Status and Tumor Necrosis Factor-Alpha Levels of Inflammatory Bowel Diseases"

_jcm, 2022, doi:10.3390/jcm11123316_

Round 1
Reviewer 1 Report
In this manuscript, the authors test the value of a model-based precision dosing (MIPD) tool for predicting infliximab trough levels, taking into account the association with disease status and circulating TNF-α in patients with inflammatory bowel disease. It is clear that monitoring infliximab levels is necessary for these patients due to considerable variability in drug exposure between patients, and MIPD tools can improve patient outcomes by guiding dose and improving disease control.
Major comments
The data is interesting, and the MIPD tool is powerful. Still, the authors need to validate the results in at least one other independent cohort to be sure that the proposed methods are reproducible.
Minor comments:
- Page 2, line 77: Add reference 11 in the correct format.
- If possible, it would be interesting to add Montreal’s classification in the patient characteristics. And test whether disease location or
- The quality of graphics should be increased. Moreover, the titles on the axes of the graphs are not displayed properly.
Author Response
- In this report we have established the performance characteristics of MIPD in forecasting trough IFX concentrations by using prior information from a model that uses two compartment pharmacokinetics with weight, ATI and albumin as covariates [as reported by Xu et al. Clin Pharmacol Drug Dev. 2012;1:203]. in this study, the parameter estimates from that prior published model were tested in this independent cohort with all specimens collected during the linear phase of elimination of IFX, and trough levels were forecasted by calculating the conditional distribution of the individual parameters. It follows that this cohort is actually a validation cohort of the published model. This as been added to the revised manuscript (line 239)
- We have followed the recommendation of the reviewer and have added the Montreal classification criteria in table 2 of the revised manuscript. Given the low number of patients in this study we respectfully believe that the analysis of disease location on forecast is underpowered.
- The quality of the graphic has been improved as presented in figure 1 and 2.
Reviewer 2 Report
In this study, Primas et al test the accuracy of model informed precision dosing tool forecasting IFX trough levels in IBD. This is overall an interesting study and gives insight into the dynamics of IFX levels during the treatment interval. It is well written and can be an asset to clinical practice.
Some minor remarks/questions
1. Is there an influence of immunomodulator use as this has been shown to increase the IFX trough levels in the SONIC trial?
2. Why do the authors refer to a 5 µg/mL threshold for the IFX trough levels although in clinical practice 3-7 is used as an optimal trough level interval?
3. Was the accuracy of the model affected by disease activity? E.g. did it perform worse in patients with active disease?
Author Response
- We have followed the recommendation of the reviewer and evaluated the impact of immunosuppressants on the PK profile, using Clearance as there were some differences in dosing. There was no significant difference in IFX Clearance between the group of patients who received concomitant immunomodulators (median 0.323 L/day, IQR: 0.295-0.385, n=8) as compared to those who did not (median 0.301 L/day, IQR: 0.234-0.435, n=33) (p=0.68).
- In our manuscript we are referring to a cutoff of 5 ug/ml which is centered around the 3 to 7 ug/ml ranges suggested (5±2 ug/ml). This choice is necessary in order to provide estimate at cutoffs, above or below the threshold
- We have followed the recommendation of the reviewer and have tested the performances of the forecast by disease activity status. The performances of the MIPD in the group of patients with (R2= 0.89; slope=0.91; n=18) or without clinical and biochemical remission status achieved (R2= 0.75; slope=0.92; n=23) is presented in the revised manuscript (line 126).
Reviewer 3 Report
Expand the number of study individual
Experimenting with dosage is experimenting on humans
Use a number of biologics need to be incorporated into the study design to preclude your group from being viewed as being n the service of a pharmaceutical company
Answer the question of why the study is needed before introducing your your methodology methodology
Author Response
We agree with the reviewer that the manuscript may benefit from additional individuals, and our group is actively analyzing additional data. This has been added to the revised manuscript (line 251). We have followed the recommendations of the reviewer and incorporate the notion that MIPD can be used to tailor therapy to other monoclonal antibodies as well as biosimilars (line 268). We have also added a statement regarding the need of the study (line 65).
Round 2
Reviewer 1 Report
This revised manuscript by Primas et al has addressed all the concerns of this reviewer. With the respective changes the authors made in their manuscript, it is now significantly improved. I believe it is ready for publication and have no further criticisms.
Reviewer 3 Report
The technology described may have future applicability for diseases whose pathogenesis is based on a persistent dysfunctional proinflammatory response to an identified antigen. In the absence of infection presence or induction, incremental adjustment of drug concentration for inflammatory bowel disease diseases stands to enhance the drug-induced clinical response over the short term. Biologics produce clinical remissions in 40% of individuals treated with tumor necrosis factor-alpha compounds.